# Understanding the Meaning of Conformity to Feminine Norms in Lifestyle Habits and Health: A Cluster Analysis

**DOI:** 10.3390/ijerph17041370

**Published:** 2020-02-20

**Authors:** Sara Esteban-Gonzalo, Petula Sik Ying Ho, Marta Evelia Aparicio-García, Laura Esteban-Gonzalo

**Affiliations:** 1Faculty of Biomedicine, Psychology Department, Universidad Europea de Madrid, 28670 Madrid, Spain; 2Department of Social Work and Social Administration, The University of Hong Kong, Hong Kong, China; psyho@hku.hk; 3Faculty of Psychology, Universidad Complutense de Madrid, 28223 Madrid, Spain; meaparic@ucm.es; 4Faculty of Biomedicine, Nursing Department, Universidad Europea de Madrid, 28670 Madrid, Spain; laura.esteban@uam.es; 5Faculty of Medicine, Nursing Department, Universidad Autónoma de Madrid, 28029 Madrid, Spain

**Keywords:** feminine role conformity, lifestyle, marital status, alcohol, tobacco

## Abstract

*Background:* Gender roles impact different spheres of life and lead women to behavioral patterns and lifestyle habits associated with femininity, generating important differences between men and women in health. The present study analyzed relationships between conformity to the feminine norms and different lifestyle indicators: Educational level, marital status, alcohol consumption, tobacco consumption, sleeping hours, social support, and physical activity. Additionally, cluster analysis was developed in order to identify different patterns of gender role conformity. *Methods:* The sample was made up of 347 women age 18–70 from Spain. Data collection was conducted during 2014. *Results:* Multiple logistic regression analyses produced odds ratios showing that women with lower feminine role conformity were more likely to use tobacco and alcohol, but less likely to share their lives with someone. Cluster analysis found four different profiles of gender role conformity related to different patterns of alcohol consumption and marital status. *Conclusions:* Conformity to feminine norms was associated with basic affective conditions such as sharing life with others and with alcohol and tobacco consumption, but not with physical activity, social support, and sleep duration. Whereas tobacco and alcohol use have important health implications, public health systems should pay attention to gender-related variables in order to design and implement specific prevention programs.

## 1. Introduction

The existence of gender roles and their effects on women’s lives has been widely investigated [1]. During recent decades, biopsychological approaches have emphasized the importance of sociocultural factors as predictors of health [2]. Gender roles are associated with patterns of behavior which, in the long term, could contribute to differences in lifestyles [3].

Gender roles and stereotypes are learned early in life, and exposure to traditional gender roles in society reinforces implicit gender beliefs such as stereotypic traits, abilities, and roles [4]. Conformity to feminine roles has been linked to characteristics such as care of others, thinness, sexual fidelity, modesty, domesticity, passivity, etc. [5,6], with significant effects on women’s self-concept and career aspirations [4,7]. Thus, conformity to feminine roles leads women to behavioral patterns and lifestyle habits associated with femininity, generating important differences between men and women in employment, education, social structure, and health [8].

In Spain, as in other countries in Europe, gender roles still have significant effects on women’s daily lives. Economic and social duties and priorities marked by gender roles could somehow generate differences in the way women use their time, the activities they engage in, and the lifestyle habits they adopt regarding healthcare [1]. 

Different roles contribute to different priorities in life, and the roles of women are strongly influenced by social context [2]. Whereas the traditional feminine role has been associated with the private sphere and its values and functions, such as the care of children and family, the traditional masculine role has been associated with the public sphere identified with work, industry, commerce, politics, and production [9]. Therefore, it is reasonable to think that women spend their time and resources on different things and activities than men, generating gender differences between men and women in lifestyles and health [10]. 

Gender differences are also observed in health. In Spain, women have reported poorer self-assessed general health, as well as stronger prevalence of psychological disorders compared to men (INE 2018). Although women have reported higher morbidity than men, data have shown that life expectancy in men is shorter than in women (INE 2018). Such findings are part of the traditional morbidity–mortality paradox: Women live longer than men do, but their health is worse [1]. Possible explanations for the shorter life expectancy among men may be related to the embracing of health-risk behaviors such as substance consumption or lower healthcare awareness among men [1]. For example, it has been found that alcohol and tobacco consumption is higher among men [11], and that these substances are traditionally associated with masculinity [12]. 

Gender may also influence health-related variables such as sleeping patterns and the use of leisure time. Self-reported sleep duration and biomedical studies have shown that women sleep more than men, which has been linked to gender differences in time use and to the amount of paid and unpaid work they do [13,14,15]. Engendered responses related to work and family roles are well known, as is their incompatibility with other leisure activities [16,17]. Similarly, gender differences in physical activity have also been explored, and men have been found to be physically active than women [18,19,20,21]. Some authors have highlighted the role of family responsibilities and higher perceived barriers to participating in physical activities as possible explanations [22]. Biological sex does not offer reasonable explanations to justify gender differences in lifestyles. Therefore, assuming a biopsychosocial perspective could help us improve our understanding of the gender gap in society and decrease the relevant existing gender disparities. Additionally, identifying patterns of gender role conformity could facilitate a more sophisticated picture of Spanish women and their gender-related habits. For this reason, the objective of this study was to analyze the associations between conformity to feminine norms and different lifestyle indicators among Spanish women. 

## 2. Method

### 2.1. Participants

The study sample was composed of 347 women. All participants were living in Spain at the time of the study, and most of them were Spanish women (93.7%), age 18–70 (mean = 42.2 years). Data collection was conducted during 2014 with the collaboration of several organizations from various sectors (healthcare, telecommunications, education, engineering, banking and insurance, business administration, and marketing and design), local employment agencies, and local women’s associations in more rural areas. These organizations offered the opportunity to their employees or associated members to voluntarily participate in this study.

The eligibility criterion was to be employed by or a member of one of the cooperating organizations. Ninety percent of women agreed to participate in the study, and around 80% of questionnaires were fully answered and their data was retained for the analyses. Missing answers were particularly observed in sexual content items among older women. A total of 86 (20%) questionnaires with missing answers were discarded. Data collection was conducted during 2014 and the data collection process was completed in approximately six months. All women received written questionnaires to answer and return to the researcher, in which they were requested to affirm that they met inclusion criteria (which was also confirmed by study staff). The average time needed to answer the questionnaire was 45–60 min after prior clarification of instructions. The participants were supported at all times by a person trained to answer their questions whenever necessary, especially older women. All participants were informed of the purpose and intent of the study and provided written consent. Similarly, anonymity of each of the participants was ensured.

### 2.2. Measurement Instruments

#### 2.2.1. Conformity to Gender Role Indicators

Conformity to the female gender role was assessed by The Conformity to Feminine Norms Inventory (CFNI), an instrument designed to measure the level of compliance of women to female standards [5]. This instrument measures the female gender role understood as the degree of women’s adherence to the rules and social standards of femininity through behaviors, feelings, and thoughts [5]. For the present study, the Spanish version of the CFNI was employed, respecting the original structure, with adequate reliability and Cronbach’s Alpha for the total scale of 0.87 [23]. The total score ranged between 0 (minimum score) and 249 (maximum score). The CFNI is composed of eight subscales whose reliability in the present study ranged between 0.64 and 0.86: Nice in relationships (reliability 0.73), care of children (0.86), thinness (0.80), sexual fidelity (0.80), modesty (0.65), romantic relationships (0.64), domestic (0.75) and invest in appearance (0.68). The total score was dichotomized into two categories: “Medium-high” and “medium-low”. The cutoff value was established at the 50th percentile (score 148.4) to divide participants into two similarly sized groups. The median split approach is considered an acceptable classification method in gender studies [24,25].

#### 2.2.2. Social Support Measures

Social support was measured by The Duke-UNC Functional Social Support Questionnaire, an instrument designed to assess an individual’s perception of the amount and type of personal social support. The Spanish version is composed of 11 items and includes quantitative and functional measures regarding affective support (the possibility of having people to communicate with) and confidant support (expression of love, affection, and empathy). The Spanish questionnaire was shown to be valid and reliable with an internal consistency for the total scale of 0.90 (Cronbach´s Alpha) and reliability coefficients between 0.80 and 0.92 [26,27]. 

#### 2.2.3. Lifestyle Indicators

Lifestyle indicators comprise variables related to sociodemographic and health characteristics: Marital status, alcohol and tobacco consumption, daily sleeping hours, and physical activity. All these variables were obtained through single questions, asking respondents to place themselves within a specific category except for age and daily sleeping hours, where continuous amounts were asked for. Marital status was categorized as alone (divorced, widowed, and single) or not alone (married or married de facto). Alcohol consumption was dichotomized as yes (those who had drunk alcohol in the last two weeks) or no (those who had not drunk alcohol in the last two weeks). Tobacco consumption was dichotomized as yes (daily consumption) or no (occasional or no consumption). 

#### 2.2.4. Covariates

Age and level of education were included as covariates to control their potential confounding effect on relations between gender conformity and lifestyle indicators. Age was used as a continuous variable. Level of education was categorized as basic (primary and secondary school) and medium-high (technical and university studies).

### 2.3. Ethical Procedures

The protocol for the present study obtained approval from the Ethics Committee of the Faculty of Psychology of the Complutense University of Madrid. All participants were informed of the purpose and intent of the study and provided written consent. Similarly, anonymity of each of the participants was ensured.

### 2.4. Data Analyses

The characteristics of the sample are described as frequencies (percentages) or mean ± SD. Statistical differences were identified by the chi-square test (categorical variables) and ANOVA test (continuous variables). Statistical significance was set at *p* > 0.05 (two-tailed).

Logistic regression has been implemented in various fields of study, particularly in research into crops [28] and livestock [29] and in the social sciences [30,31,32]. Binary logistic regression models were constructed to analyze the relationships between gender role conformity and civil status, alcohol consumption, tobacco consumption, and physical activity, controlling for confounding variables (age and educational level). In addition, linear regression models were built to assess the relationship between gender role conformity and sleep duration and social support.

Three regression models were built with medium-high conformity women as a reference group. Model 0 was crude, Model 1 was adjusted for age, and Model 2 was additionally adjusted for level of education. The results of the models were presented as adjusted odds ratios (OR) with their 95% confidence intervals (CI). The statistical power range was 0.91–0.95, calculated to detect risk differences of 1.5 or higher.

In addition, cluster analysis was carried out to identify patterns of gender role conformity based on information provided by the eight subscales of the CFNI. This procedure attempts to identify relatively homogeneous groups of cases based on selected characteristics. Hierarchical clustering using Ward’s method was applied using standardized subscale scores and Euclidean distances. Dendrograms were created during the clustering process to help determine how many clusters should be included in the final solution. Cluster consistency was assessed using ANOVA analysis. 

Finally, patterns of gender role conformity (identified by the cluster analysis) were related to those lifestyle variables previously found to be linked to gender role conformity (marital status, tobacco and alcohol use). The chi-square test was used for this analysis.

## 3. Results

### 3.1. Descriptive Analysis

Differences between women with medium-low and medium-high conformity to feminine norms are presented in Table 1. Differences between both groups in marital status were significant (*p* = 0.003). The percentage of divorced, widowed and single women was higher in the medium-low conformity group than in the medium-high conformity group. Accordingly, the percentage of married and married *de facto* women was higher in the medium-high conformity group than in the medium-low conformity group. Similarly, the chi-square test indicated significant differences (*p* = 0.005) in alcohol consumption between both groups with a higher frequency of alcohol consumption among medium-low conformity women than among medium-high conformity women. The percentage of daily smoking women was also significantly (*p* = 0.041) higher among medium-low conformity women than among medium-high conformity women. 

### 3.2. Multivariate Analysis

The relationship between conformity to feminine norms and lifestyle indicators are presented in Table 2. Medium-high conformity women were more likely to live with someone (married and married *de facto* women) than medium-low conformity women (OR = 1.89; 95% CI = 1.23–2.89; *p* = 0.003). Results were similar in Model 1 after adjusting for age (OR = 1.80; 95% CI = 1.13–2.85; *p* = 0.012), and Model 2 after adjusting for age and educational level (OR = 1.70; 95% CI = 1.06–2.71; *p* = 0.025).

Medium-high conformity women were less likely to drink alcohol than medium-low conformity women (Model 0) (OR = 1.84; 95% CI = 1.20–2.82; *p* = 0.005). Results were similar in Model 1 after adjusting for age (OR = 1.77; 95% CI = 1.14–2.72; *p* = 0.010) and Model 2 after adjusting for age and educational level (OR = 1.65; 95% CI = 1.06–2.56; *p* = 0.026).

Medium-low conformity women showed a higher likelihood of consuming tobacco than medium-high conformity women (Model 0) (OR = 1.75; 95% CI = 1.02–3.00; *p* = 0.042). However, differences were no longer significant after controlling for age in Model 1 (OR = 1.64; 95% CI = 0.95–2.84; *p* = 0.076) and for age and educational level in Model 2 (OR = 1.70; 95% CI = 0.98–2.96; *p* = 0.059).

There were no significant differences regarding conformity to feminine roles and physical activity (Table 2), sleep duration, and social support in any of the three models constructed (data not shown in table: β = −0.24, 0.15 (SE), *p* = 0.103 for Model 1; β = −0.27, 0.14 (SE), *p* = 0.068 for Model 2; β = −0.24, 0.15 (SE), *p* = 0.103 for Model 3).

### 3.3. Cluster Analysis and Lifestyle Indicators

Cluster analysis suggested four different patterns of gender role conformity. A four-cluster solution seemed the most adequate and feasible grouping combination, highlighting differences between groups in a meaningful way. The scores of each group (mean scores and standard deviations) are presented in Table 3 and Figure 1. The associations between conformity to gender roles and lifestyle indicators are presented in Table 4.

Group 1 (the moderate group) consisted of 99 women and was characterized by a medium gender role conformity in most of the CFNI subscales, except for a higher than average score in involvement with children and a much lower than average score in nice in relationships. This group showed standard average patterns of alcohol use, and the proportion of married and married *de facto* women was slightly higher than the average. This group of women was also older than average and reported a smaller proportion of medium-high studies. 

Group 2 (the conformist group) was the largest with 126 women and was characterized by a high gender role conformity in most of the CFNI subscales, except for subscales of thinness, romantic in relationships, and investment in appearance, in which the scores were medium. This group showed lower rates of alcohol use and higher rates of married and *de facto* married women compared to other groups. This group of women were the oldest among all the groups and also reported the lower proportion of medium-high studies. 

Group 3 (the contrast group), with 85 women, was the most divided group in terms of conformity. Scores in subscales nice in relationships, thinness, romantic, domestic, and investment in appearance were noticeably higher than average, and scores in subscales involvement in children, sexual fidelity, and modesty were significantly lower than average. Domestic seemed to be the only medium score. In this group, alcohol consumption was much higher than average, as was the proportion of divorced, single, or widowed women. This group of women were younger than Groups 1 and 2 and reported a higher than average proportion of medium-high studies. Last, Group 4 (the nonconformist group), the smallest with just 37 women, was defined by low gender conformity. All CFNI subscales showed low scores. This group showed a higher proportion of divorced, single, or widowed women but average rates of alcohol use. The women in Group 4 were the youngest of all and reported the highest proportion of medium-high studies. 

## 4. Discussion

As expected, our results showed that gender roles were connected with some lifestyle behaviors, but not with others. Conformity to feminine norms was associated with basic affective conditions such as sharing life with others (e.g., marital status) and with alcohol and tobacco consumption, but not with physical activity, social support, or sleep duration. 

Women with lower gender role conformity are significantly more likely to live alone, which, in this research, was understood as being divorced, widowed, or single. Some authors have pointed out that marriage contributes to gender inequality in terms of power and authority in both the family and society [33], stressing the link between marriage and male domination [34]. Family roles and childcare have been identified as potential barriers to career success [35], while also affecting women’s economic empowerment [7] and therefore influencing gender inequality. The number of single-parent families has increased over recent decades in Spain (EPA, 2018), which, in other countries, single-parent families have been associated with an increased need to participate in remunerated work to support their families and themselves [35]. These circumstances generate a new social context for women in which life demands exceed those traditionally assigned to women [2,36,37,38]. In the 20th century and in the past, marriage has represented the only socially accepted way to include women in the Spanish society, denying them any type of individual independence and relegating women to subordinate positions in society [39,40]. However, Spanish women’s roles within the household is progressively changing, incorporating different family models (mono-parental and same-sex families) in which less conformist women feel more identified [41]. 

In addition, our results showed that women with lower gender role conformity presented a higher likelihood of using tobacco and drinking alcohol. Tobacco and alcohol consumption have been traditionally labeled as masculine habits [1]. Males report greater use of both substances than females [42], and this behavior is hypothesized as an explanation of important health-risk behaviors and thus differences in longevity between men and women [43]. The use of tobacco and alcohol has been traditionally linked to public and social life, and usually more restricted to women [44]. Although recent tendencies may show changes, our results found that alcohol and tobacco use are still connected to gender roles, at least in Spain, where similar studies found that several feminine norms, such as romantic relationships and sexual fidelity were inversely correlated with alcohol and cigarette consumption, and investment in appearance was inversely related to tobacco consumption [1]. In Spain, alcohol and tobacco use has not only been traditionally reserved for men, but also linked to parties and social events where women were usually excluded. For that reason, tobacco and alcohol use has become a mean of making equal gender behaviors, representing women’s conflicts between liberal and traditional social norms [45,46]. However, results were no longer significant in the case of tobacco consumption after adjusting for age and educational level. These results highlight the importance of age and educational level as important moderators between gender roles and tobacco consumption, as has been previously found in other studies linking gender roles and health, suggesting interactions between different social variables. 

No significant associations of daily sleeping hours, social support, and physical activity with gender role conformity were found. Although other studies have found differences in sleep durations and sleeping habits between men and women, with a higher amount of sleeping hours among females [13,14,15], these results were not observed in our study. Similarly, our findings are not aligned with the existing literature in the case of physical activity, which was expected to be related to gender role conformity. Differences in leisure activities and use of time have been associated with gender roles [16,17], and work and family roles have been perceived as possible barriers between women and outdoor physical activities [18,19,21,22]. Nevertheless, self-care and body image are also associated with certain gender roles, such as investment in appearance [5], which could somehow contribute to encouraging women to engage in sports and physical activities. During the second half of the 20th century, the repression of women in Spain had confined them within domestic spaces in which households and nonremunerated responsibilities (cleaning house, care of children, etc.) were women’s priorities. Therefore, self-care and women’s needs were invisible. Sports practice and outdoor activities used to be considered masculine activities [47,48]. Nevertheless, over recent decades, social pressure toward women in terms of body image has grown dramatically with the ideal of the perfect body, forcing women to stay thin, beautiful, and young [5]. However, none of these effects were observed in our results. These results might help to dispel some myths among Spanish women in a social context in which even the more conformist women may need to redefine some lifestyle behaviors to adapt to new social demands. 

Another relevant aspect to be discussed is the lack of effect of age and educational level on the relationship between conformity to feminine norms and lifestyle behaviors, with the exception of tobacco consumption. Prior studies and hypotheses have claimed that educational level may somehow contribute to moderating or attenuating gender roles [49]. Women with a higher educational level have been found to be less conditioned by gender roles, placing them in more tolerant and egalitarian positions [7]. However, these results were not found in this study. Possible explanations to justify these discrepancies could be that the relation between both variables is not easily evident or visible, but more sophisticated. 

Finally, no significant relations were found between feminine role conformity and any of the social support indicators (affective support, confidential support, and total score). Previous studies have shown contradictory data regarding support, but most of the existing literature has pointed out that high feminine role conformity would be positively related to social support in terms of either seeking or receiving social support [50,51,52]. However, our results do not allow us to support this hypothesis. 

### Patterns of Femininity and Lifestyles

Cluster analysis helped us to describe and understand different patterns of femininity, which reflect how women adapt their lifestyles to social standards in order to fulfill both social and individual needs.

Group 1, the moderate group representing almost a third of the sample, shows a significant number of women at medium levels of gender role conformity who still seem to focus their attention on childcare. This phenomenon could be linked to the fact that childcare is still strongly delegated to women in Spain [53,54], even in less traditional households in which both members contribute actively to the family economy [55,56,57,58,59]. Although gender roles within households are changing in Spain, with more alternative family models appearing, care of children might be one of the most change-resistant aspects among gender roles. Results suggest that even among moderately conformist women, care of children is still a considerable life choice. The fact that this group of women showed lower presence of medium-high studies could be aligned with previous studies that pointed out education as a way to attenuate gender roles [60]. In addition, the higher age observed when compared to more liberal groups (the contrast and the nonconformist group) may suggest the visualization of the gender gap across generations [61]. Finally, the higher rate of sharing life with someone in this group could be connected with their inclination to the care of children in terms of showing a higher tendency to group in family nuclei.

Group 2, the conformist group, seems to represent a considerable number of Spanish women that still identify with the traditional standards of femininity, but are somewhat more relaxed in terms of external appearance. Results coincide with other descriptive findings, in which a high proportion of medium-high conformity has been observed [23]. In Spain, as in other southern European countries, gender roles still have significant influence in society [57,62], with little evolution in gender stereotypes over recent decades [63]. This finding appears to be reflected in certain patterns of substance use (alcohol, according to our data) and the way they share their lives with others focused on traditional household structure, reinforcing the previously explained link between marriage and alcohol use with gender role conformity. This group of women was also older and had a lower educational level than the average woman, again supporting the preventive effects of education against gender inequalities [64]. It makes visible a large group of women at a higher age who identify with traditional feminine roles as part of their lifestyles. Moreover, although feminine role conformity has been found to damage women’s psychological health [60], conformity may actually be a protective factor in terms of keeping them away from unhealthy traditionally masculine-related practices such as alcohol use. 

Group 3, the contrast group, represents almost a fourth of the sample and is defined by contrast in the CFNI subscales: High gender role conformity in most of the subscales except in involvement in children, sexual fidelity and modesty, and a medium score in domesticity. The group presents an external image adapted to femininity standards, but is more liberal in terms of sexual freedom, less focused on children, and less interested in showing their abilities.

Possible explanations for these findings could be related with the growing sector of Spanish women engaged in highly demanding professions, which forces them to choose between involvement in family and career aspirations [65,66,67,68], thus developing more egalitarian gender roles and work-related attitudes and behaviors. This fact was observed in their patterns of alcohol use, which were more connected with the highly demanding professional and public life and less conservative households in which single or divorced women are more common [1,35,42,44]. Nevertheless, social pressure in terms of physical appearance and kindness toward others is still a gender-related variable even among the most demanding and competitive job positions, at least in the Spanish context [69,70]. This hypothesis seems to be supported by the fact that a higher proportion of medium-high studies was found in Group 3, representing younger Spanish women who, despite conforming to some gender roles, still try to compete in an increasingly demanding working environment. The same environment which demands feminine standards, such as being nice to others and preserving a great external appearance, also allows them to practice more liberal ideas and principles in their private life. They preserve their freedom and choose to preserve their autonomy. They choose to stay single and believe in sexual freedom more frequently than conservative groups. 

Finally, Group 4, the nonconformist group, represents a small percentage of women who dismiss the social standards of feminine role conformity. These results coincide with other authors’ findings that only a low proportion of women show low gender role conformity patterns [5,23]. Unexpectedly, a higher alcohol use was not reported among this group of women. Nevertheless, a higher presence of divorced, single or widowed women was found, concurring with the idea of marriage as the reinforcement of gender inequalities in social structures [33,71,72]. This group of women, the youngest and with the highest educational level, seems to represent a growing sector of women that is more aware of gender inequalities. They struggle for their rights by adopting more equalitarian gender-related behaviors. Regarding alcohol use, previously linked to masculine habits [1], it could be speculated that these women do not need to adopt masculine habits to struggle for they rights.

## 5. Conclusions

In conclusion, it seems reasonable to consider that feminine role conformity has an influence on some lifestyle habits. According to our results, gender role conformity may influence the way women decide to share their life with others and their substance consumption behavioral patterns with respect to alcohol and tobacco. The reasons why people establish priorities in life are extremely complex, but in any case, gender roles should be included as one more variable in order to understand such complexity. Whereas tobacco and alcohol use have important health implications, public health systems should pay attention to gender-related variables in order to design and implement specific prevention programs. Moreover, the fact that gender roles evolution may redefine our society in terms of social structures with significant implications in household’s dynamics, distribution and needs is equally important.

### Limitations and Future Research

This study was not without its limitations. First, the scope of the sample should be improved in future research because our relatively modest sample size may have provided inadequate statistical power to detect slight variations as statistically significant. Second, our sample was not representative of the population of Spanish women, which reduces the generalizability of our results. Third, given the limited sample size employed in the analysis performed, we were not able to detect risk differences lower than 1.5. Fourth, measurements of alcohol and tobacco use could be improved in future studies by adding the actual amount of ingested alcohol and smoked cigarettes to frequency of use. Finally, the cross-sectional design of our study prevents determination of the temporal relation of the variables and thus the detection of cause-effect relationships. 

## Figures and Tables

**Figure 1 ijerph-17-01370-f001:**
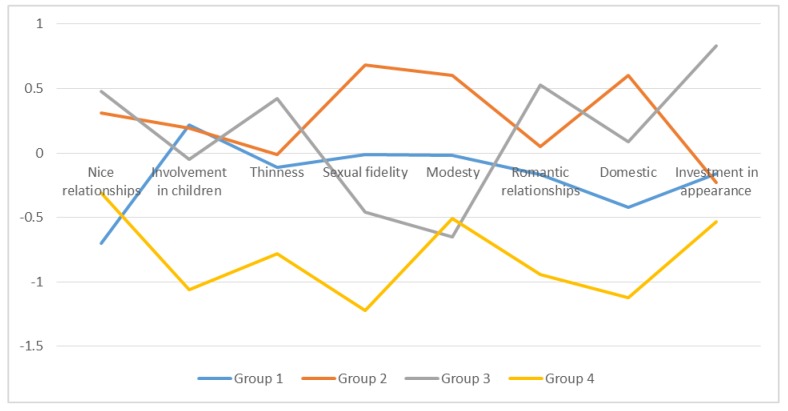
Patterns of feminine role conformity (* standardized scores).

**Table 1 ijerph-17-01370-t001:** Characteristics of the study sample by conformity to feminine roles.

Lifestyle Variables	Total	Medium-Low Conformity	Medium-High Conformity	*p*
	(*n* = 347)	(*n* = 178)	(*n* = 169)	
*Age* (*mean*, *SD*)	42.2 (14.0)	41.0 (14.0)	43.4 (13.9)	0.113 ^a^
*Educational Level*				
Basic Studies (%)	19.9	16.3	23.7	0.085 ^b^
Medium-High Studies (%)	80.1	86.7	76.3	
*Marital Status*				
Alone (%)	51.3	59.0	43.2	**0.003** ^b^
Not alone (%)	48.4	41.0	56.8	
*Alcohol Consumption* (*2 weeks*)				
Yes (%)	52.7	60.1	45.0	**0.005** ^b^
No (%)	47.3	39.9	55.0	
*Tobacco Consumption*				
Yes (%)	19.9	24.2	15.4	**0.041** ^b^
No (%)	80.1	75.8	84.6	
*Daily sleeping hours* (*mean*, *SD*)	7.39	7.48	7.29	0.113 ^a^
*Social Support* (*mean*, *SD*)				
Affective Support	18.72 (4.4)	18.81 (4.3)	18.64 (4.5)	0.675 ^a^
Confidential Support	23.98 (5.5)	24.12 (5.2)	23.84 (5.8)	0.640 ^a^
Total Score	42.58 (9.5)	42.81 (9.1)	42.34 (9.9)	0.641 ^a^
*Physical Activity*				
Yes (%)	67.4	68.5	66.3	0.652 ^b^
No (%)	32.6	31.5	33.7	

^a^ ANOVA; ^b^ Pearson chi-square. *p* < 0.05 in bold.

**Table 2 ijerph-17-01370-t002:** Logistic regression models for the association between The Conformity to Feminine Norms Inventor (CFNI) and lifestyle variables.

Lifestyle Variables		Model 0			Model 1			Model 2	
	OR	95% CI	*p*	OR	95% CI	*p*	OR	95% CI	*p*
*Marital Status*	1.89	1.23–2.89	**0.003**	1.80	1.13–2.85	**0.012**	1.70	1.06–2.71	**0.025**
*Alcohol Consumption*	1.84	1.20–2.82	**0.005**	1.77	1.14–2.72	**0.010**	1.65	1.06–2.56	**0.026**
*Tobacco Consumption*	1.75	1.02–3.00	**0.042**	1.64	0.95–2.84	0.076	1.70	0.98–2.96	0.059
*Physical activity*	0.96	0.56–1.65	0.888	0.92	0.53–1.59	0.775	1.01	0.58–1,78	0.952
*Age*	1.01	0.99–1.02	0.113						
*Educational level*	0.698	0.53–0.91	**0.010**						

Model 1, adjusted for age; Model 2, adjusted for age and educational level. *p* < 0.05 in bold.

**Table 3 ijerph-17-01370-t003:** Characteristics of the formed groups by CFNI subscales.

CFNI Sub-Scales (*Mean*, *SD*)	Group 1 (*n* = 99)	Group 2 (*n* = 126)	Group 3 (*n* = 85)	Group 4 (*n* = 37)	Total (*Mean*, *SD*)	*p* *
Nice in Relationships	32.5 (3.1)	38.0 (5.3)	38.9 (5.2)	34.6 (4.8)	36.3 (5.4)	<0.001
Involvement in Children	22.7 (3.8)	22.5 (7.3)	20.8 (6.6)	14.1 (6.4)	21.2 (6.7)	<0.001
Thinness	16.5 (3.4)	16.8 (6.1)	19.3 (6.7)	12.4 (4.2)	16.9 (5.7)	<0.001
Sexual Fidelity	17.15 (3.3)	21.2 (5.3)	14.5 (5.0)	10.1 (4.4)	17.2 (5.8)	<0.001
Modesty	13.5 (1.9)	15.7 (2.9)	11.3 (3.9)	11.8 (3.9)	13.6 (3.5)	<0.001
Romantic Relationships	12.7 (2.2)	13.5 (3.5)	15.1 (3.5)	10.1 (3.0)	13.3 (3.4)	<0.001
Domestic	16.1 (2.1)	19.5 (2.9)	17.8 (3.3)	13.8 (2.6)	17.5 (3.3)	<0.001
Investment in Appearance	11.5 (2.0)	11.3 (3.3)	14.5 (2.5)	10.4 (2.7)	12.0 (3.0)	<0.001

* ANOVA test.

**Table 4 ijerph-17-01370-t004:** Patterns of feminine role conformity and lifestyles indicators.

Lifestyle Variables	Group 1	Group 2	Group 3	Group 4	Total	*p*
	(*n* = 99)	(*n* = 126)	(*n* = 85)	(*n* = 37)	(*n* = 347)	
*Age* (*mean*)	44.25+	47.42+	35.48−	34.24−	42.2	**<0.001** ^a^
*Educational Level*						
Basic Studies (%)	26.3+	28.6+	5.9−	5.4−	19.9	**<0.001** ^b^
Medium-High Studies (%)	73.7−	71.4−	94.1+	94.6+	80.1	
*Marital Status*						
Alone (%)	42.4−	38.1−	74.1+	67.6+	51.3	**<0.001** ^b^
Not alone (%)	57.6+	61.9+	25.9−	32.4−	48.4	
*Alcohol Consumption* (*2 weeks*)						
Yes (%)	52.5	38.1−	71.8+	49.5−	52.7	**<0.001** ^b^
No (%)	47.5	61.9+	28.2−	50.5+	47.3	
*Tobacco Consumption*						
Yes (%)	19.2	18.3	21.2	24.3	19.9	0.853 ^b^
No (%)	80.8	81.7	78.8	75.7	80.1	
*Daily sleeping hours* (*mean*)	7.4	7.24	7.51	7.59	7.39	0.224 ^a^
*Social Support* (*mean*)						
Affective Support	18.58	18.55	18.68	19.76	18.72	0.505 ^a^
Confidential Support	23.73	23.47	24.25	25.81	23.98	0.135 ^a^
Total Score	42.3	41.94	42.69	45.27	42.58	0.304 ^a^
*Physical Activity*						
Yes (%)	71.7	65.9	61.2	75.7	67.4	0.305 ^a^
No (%)	28.3	34.1	38.8	24.3	32.6	

^a^ ANOVA; ^b^ Pearson chi-square; *p* < 0.05 in bold.

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
