# Peer review of "Understanding the Meaning of Conformity to Feminine Norms in Lifestyle Habits and Health: A Cluster Analysis"

_ijerph, 2020, doi:10.3390/ijerph17041370_

Round 1
Reviewer 1 Report
Thank you for revising the manuscript, you have addressed all of my questions and concern and I believe the manuscript is ready to publish. Good job!
Reviewer 2 Report
Authors have addressed all comments satisfactory and now paper is publishable in the present form.
Reviewer 3 Report
I am satisfied with the authors' responses to my original comments.
This manuscript is a resubmission of an earlier submission. The following is a list of the peer review reports and author responses from that submission.
Round 1
Reviewer 1 Report
Comments:
The second part of the following sentence is against the first section, please rephrase:“Gender differences are also observed in health. In Spain, women have reported poorer self-assessed general health, as well as stronger prevalence of psychological disorders compared to men (INE 2018)”.
2) Please clarify in the method:
How authors computed the sample size? add the formula for that. What was the Response rate? Time period of study?
3) Explain more about CFNI. What is the max score, 8? and minimum, zero?
4) In Table 1; please use another symbol to address the ANOVA and chi-sq for example + and $ etc., * usually used for significantly level and generates some confusion about the significant level
5) My suggestion is to add a table and compare G1-G2, G3 and G4 for 8 elements and show the + and - signs for greater and slower, at the end the comparison will present which elements repeated + and - across four groups.
6) p.4: what was the cut point to define medium-high conformity women as a reference group?
7) T2, can you add OR for age and education level, it may explain the changes in OR between 3 models, also add N and AIC.
8) I was thinking to add the T1 components to T3, the analysis and message of the article are round about the cluster analysis but T3 does not provide enough information to think about the sample characteristics, in that case authors can modify the page 7 (lines 196-214 and pp8-9: 282-317) to discuss more about the sample characteristics.
9) I expected to see the "sexual fidelity" in G3, please check it again. What happened to G4, the combination of sample does not make any sense, authors may need to force the model to 3 groups instead of 4 (optional).
10)pp8-9: 282-317; This is not a real discussion a reader likes to see why this pattern happened? and how the study can explain that? please look at my comments 7 and 8 and then modify this section moderately.
I was thinking women are using Tobacco and Alcohol as a substitute for being out of community, probably they are using this as a substitute for loneliness, etc., this is the important findings and need to be focused more that is one of the reason they tend to not be physically active.
Author Response
Dr. Sara Esteban Gonzalo
Associate Professor
Faculty of Biomedical and Health Science
Universidad Europea
Villaviciosa de Odón, 28670 Madrid
Sara.esteban@universidadeuropea.es
05 February 2020
Dear Reviewer
RE submission of manuscript: No. ijerph-684541
Thank-you for giving us the opportunity to re-submit to International Journal of Environmental Research and Public Health a revised version of the manuscript entitled " Understanding the meaning of conformity to feminine norms in lifestyle habits and health: A cluster analysis.” by Esteban et al. The authors thank the reviewer for their thoughtful comments and suggestions. We have been able to incorporate all of the feedback and feel that the manuscript is stronger as a result.
All the authors have read and approved the final manuscript.
We look forward to your ongoing correspondence regarding our submission.
Sincerely,
Dr S.E.
Corresponding Author
REVIEWERS' COMMENTS
Reviewer 1.
1) The second part of the following sentence is against the first section, please rephrase:
Thank you for this interesting contribution. We have rephrase the middle part of the paragraph so that it is easier to understand the information, we hope that it is easier to understand already:
Gender differences are also observed in health. In Spain, women have reported poorer self-assessed general health, as well as stronger prevalence of psychological disorders compared to men (INE 2018). Contrary to the fact that women reported higher morbidity than men did, data show that life expectancy in men is shorter than in women (INE 2018). Such findings are part of the traditional morbidity–mortality paradox: women live longer than men do, but their health is worse (Sánchez–López, Cuellar–Flores, & Dresch, 2012). Possible explanations for the shorter life expectancy among men may be related to the embracing of health-risk behaviors such as substance consumption, or lower health care awareness among men (Sánchez–López, et al., 2012). For example, it has been found that alcohol and tobacco consumption is higher among men (Organization & Unit, 2014), and that these substances are traditionally associated with masculinity (Mahalik, Burns, & Syzdek, 2007).
2) Please clarify in the method:
How authors computed the sample size? add the formula for that. What was the Response rate? Time period of study?
Thank you for this relevant comment. We calculated the statistical power for all analysis performed employing the BPower tool, considering a minimum OR to be detected of a 1.5, the R2 (Nagelkerke) showed in the adjusted regression models built and the sample size for each analysis. We have obtained a minimum statistical power of 0,91 (considering a minimum sample size, n=290) and a maximum of 0.95 (considering the maximum sample size, n=336). However, we are aware of that we are able to detect a minimum difference of an OR=1.5 or higher. For that reason, we have included this information as a limitation (a), as well as the statistical power values in the method section (b):
Third, given the limited sample size employed in the analysis performed, we are not able to detect risk differences lower than 1.5. Three regression models were built with medium-high conformity women as a reference group. 136 Model 0 was crude, model 1 was adjusted for age, and model 2 was additionally adjusted for level of 137 education. The results of the models were presented as adjusted odds ratios (OR) with their 95% 138 confidence intervals (CI). The statistical power range was 0.91- 0.95, calculated to detect risk differences of 1.5 or higher.
Regarding the response rate and the period of study, the following information has been added to clarify both aspects; thank you for this appreciation:
Ninety percent of women agreed to participate in the study, and around 80% of questionnaires were fully answered and their data retained for the analyses. Missing answers were particularly observed in sexual content items among older women. A total of 86 (20%) questionnaires with missing answers were discarded. Data collection was conducted during 2014 and data collection process was completed in approximately 6 months.
3) Explain more about CFNI. What is the max score, 8? and minimum, zero?
Thank you for advising us about this missing information. We immediately proceed to complete it:
The total score rages between 0 (minimum score) and 249 (maximum score).
4) In Table 1; please use another symbol to address the ANOVA and chi-sq for example + and $ etc., * usually used for significantly level and generates some confusion about the significant level
Thank you for sharing this detail that could definitely contribute to make Table 1 feel more intuitive. You can find the changes operated in Table 1 within the document.
5) My suggestion is to add a table and compare G1-G2, G3 and G4 for 8 elements and show the + and - signs for greater and slower, at the end the comparison will present which elements repeated + and - across four groups.
Thank you so much for this precious suggestion. We have delated the previous table and added a new one in which all 8 elements have been connected to G1, G2, G3 and G4. Additionally, + and – signs have been added to the four group scores (only among significant associations) so that the table and its interpretation becomes more intuitive. Additionally and in line with comment 7, 8 and 10, more information has been added, both in the results, and in the discussion sections. This is a very important contribution as it helped us to see differences in age and educational level between groups, differences that we did not expect because descriptive analysis did not point it out.
Table 4. Patterns of feminine role conformity and Lifestyles Indicators.
|
|
Group 1 |
Group 2 |
Group 3 |
Group 4 |
Total |
P |
|
|
(n=99) |
(n=126) |
(n=85) |
(n=37) |
(n=347) |
|
|
Age (mean) |
44.25+ |
47.42+ |
35.48- |
34.24- |
42.2 |
<0.001ᵃ |
|
Educational Level |
|
|
|
|
|
|
|
Basic Studies (%) |
26.3+ |
28.6+ |
5.9- |
5.4- |
19.9 |
<0.001ᵇ |
|
Medium-High Studies (%) |
73.7- |
71.4- |
94.1+ |
94.6+ |
80.1 |
|
|
Marital Status |
|
|
|
|
|
|
|
Alone (%) |
42.4- |
38.1- |
74.1+ |
67.6+ |
51.3 |
<0.001ᵇ |
|
Not alone (%) |
57.6+ |
61.9+ |
25.9- |
32.4- |
48.4 |
|
|
Alcohol Consumption (2 weeks) |
|
|
|
|
|
|
|
Yes (%) |
52.5 |
38.1- |
71.8+ |
49.5- |
52.7 |
<0.001ᵇ |
|
No (%) |
47.5 |
61.9+ |
28.2- |
50.5+ |
47.3 |
|
|
Tobacco Consumption |
|
|
|
|
|
|
|
Yes (%) |
19.2 |
18.3 |
21.2 |
24.3 |
19.9 |
0.853ᵇ |
|
No (%) |
80.8 |
81.7 |
78.8 |
75.7 |
80.1 |
|
|
Daily sleeping hours (mean) |
7.4 |
7.24 |
7.51 |
7.59 |
7.39 |
0.224ᵃ |
|
Social Support (mean) |
|
|
|
|
|
|
|
Affective Support |
18.58 |
18.55 |
18.68 |
19.76 |
18.72 |
0.505ᵃ |
|
Confidential Support |
23.73 |
23.47 |
24.25 |
25.81 |
23.98 |
0.135ᵃ |
|
Total Score |
42.3 |
41.94 |
42.69 |
45.27 |
42.58 |
0.304ᵃ |
|
Physical Activity |
|
|
|
|
|
|
|
Yes (%) |
71.7 |
65.9 |
61.2 |
75.7 |
67.4 |
0.305ᵃ |
|
No (%) |
28.3 |
34.1 |
38.8 |
24.3 |
32.6 |
|
|
|
|
|
|
|
|
|
ᵃANOVA; ᵇPearson Chi-Square.
6) p.4: what was the cut point to define medium-high conformity women as a reference group?
Thank you for reminding us to include this precious information. You can find the added information below:
The cut-off value was established at the 50th percentile (score 148.4) to divide participants into two similarly-sized groups.
7) T2, can you add OR for age and education level, it may explain the changes in OR between 3 models, also add N and AIC.
Thank you for this contribution. OR for age and educational level as well as n (347 women) have been included in T2. No significant differences were obtained for conformity to femininity gender norms and age. However, there are significant differences regarding conformity to femininity gender norms and level of education.
Table 2. Logistic Regression Models for the association between CFNI and Lifestyle Variables.
|
|
|
Model 0 |
|
|
Model 1 |
|
|
Model 2 |
|
|
|
OR |
95% CI |
P |
OR |
95%CI |
P |
OR |
95% |
P |
|
|
|
|
|
|
|
|
|
|
|
|
Marital Status |
1.89 |
1.23-2.89 |
0.003 |
1.80 |
1.13-2.85 |
0.012 |
1.70 |
1.06-2.71 |
0.025 |
|
Alcohol Consumption |
1.84 |
1.20-2.82 |
0.005 |
1.77 |
1.14-2.72 |
0.010 |
1.65 |
1.06-2.56 |
0.026 |
|
Tobacco Consumption |
1.75 |
1.02-3.00 |
0.042 |
1.64 |
0.95-2.84 |
0.076 |
1.70 |
0.98-2.96 |
0.059 |
|
Physical activity |
0.96 |
0.56-1.65 |
0.888 |
0.92 |
0.53-1.59 |
0.775 |
1.01 |
0.58-1,78 |
0.952 |
|
Age |
1.01 |
0.99-1.02 |
0.113 |
|
|
|
|
|
|
|
Educational level |
0.698 |
0.53-0.91 |
0.010 |
|
|
|
|
|
|
Model 1, adjusted for age; Model 2, adjusted for age and educational level.
8) I was thinking to add the T1 components to T3, the analysis and message of the article are round about the cluster analysis but T3 does not provide enough information to think about the sample characteristics, in that case authors can modify the page 7 (lines 196-214 and pp8-9: 282-317) to discuss more about the sample characteristics.
Thank you for this observation. Despite the information of T1 and T3 is too extensive to summarize it in just one table, we hope that the changes provided in comment 5) (the new table) and the information added both in the results, and the discussion, could clarify this point, making the sample and groups’ characteristics more accessible and intuitive for the reader. (See the end of the document)
9) I expected to see the "sexual fidelity" in G3, please check it again. What happened to G4, the combination of sample does not make any sense, authors may need to force the model to 3 groups instead of 4 (optional).
Thank you for this suggestion. Actually, we had seriously considered to force the model to 3 groups, but in that case group 4 (the non-conformist group) became invisible. For us, group 4 is important as it represents a small, but growing group of women in the Spanish society who is aware of gender inequalities and therefore adopt a non-conformist perspective. This group represents all those women who struggle for their rights, adopting more equalitarian gender related behaviours. We have clarified this aspect along the discussion. We hope that is better now.
Regarding sexual fidelity in G3 (the contrast group), we have checked it twice and the score remains low. This group of women presented low scores in sexual fidelity, which make sense if you take into account that this group of women seems to be more liberal in some points. Again, we have clarified this aspect along the discussion section.
(See the end of the document)
10)pp8-9: 282-317; This is not a real discussion a reader likes to see why this pattern happened? and how the study can explain that? please look at my comments 7 and 8 and then modify this section moderately.
I was thinking women are using Tobacco and Alcohol as a substitute for being out of community, probably they are using this as a substitute for loneliness, etc., this is the important findings and need to be focused more that is one of the reason they tend to not be physically active.
Thank you for this last comment. We have definitely improved the discussion, offering more explanations and trying to make the results more understandable and logical for the reader.
4.1. Patterns of Femininity and Lifestyles
Cluster analysis helped us to describe and understand different patterns of femininity, which reflect how women adapt their lifestyles to social standards in order to fulfill both social and individual needs.
Group 1, the moderate group representing almost a third of the sample, shows a significant number of women at medium levels of gender role conformity who still seem to focus their attention on childcare. This phenomenon could be linked to the fact that care of children is still strongly delegated to women in Spain (Artázcoz et al., 2001; Artazcoz, Escribà-Agüir, & Cortès, 2004), even in less traditional households in which both members contribute actively to the family economy (Carrasco & Recio, 2001; Flaquer, Mínguez, & López, 2016; Gracia, 2014; Guner, Kaya, & Sánchez-Marcos, 2014; Mínguez, 2010). Despite gender roles within households are changing in Spain, with more alternative family models appearing, care of children might be one of the most change-resistant aspects among gender roles. Results suggest that even among moderately conformist women, care of children is still a considerable life choice. The fact that this group of women showed lower presence of medium-high studies could be aligned with previous studies that pointed out education as a way to attenuate gender roles (Esteban-Gonzalo, Aparicio, & Estaban-Gonzalo, 2018). In addition, the higher age observed when compared to more liberal groups (the contrast and the non-conformist group) may suggest the visualization of the gender gap across generations(Scott, 2006). Finally, the higher rate of sharing life with someone in this group could be connected with their inclination to the care of children, in terms of showing a higher tendency to group in family nuclei.
Group 2, the conformist group, seems to represent a considerable number of Spanish women that still feel identified with the traditional standards of femininity, but are somewhat more relaxed in terms of external appearance. Results coincide with other descriptive findings in which a high proportion of medium-high conformity has been observed (Sánchez-López, Flores, Dresch, & Aparicio-Garcia, 2009). In Spain, as in other southern European countries, gender roles still have significant representation in society (González, Jurado, & Naldini, 2014; Mínguez, 2010), with little evolution in gender stereotypes over recent decades (López-Sáez, Morales, & Lisbona, 2008). This finding appears to be reflected in certain patterns of substance use (alcohol, according to our data) and the way they share their lives with others focused on traditional household structure, reinforcing the previously explained link between marriage and alcohol use with gender role conformity. This group of women was also older and had a lower educational level than the average, supporting once again the preventive effects of education against gender inequalities (Borgonovi, 2012). It makes visible a numerous population group of women of higher age, identified with traditional feminine roles as part of their lifestyles. Moreover, although feminine role conformity has been found to damage women’s psychological health(Esteban-Gonzalo, et al., 2018), conformity may actually be a protective factor in terms of keeping them away from unhealthy traditionally masculine-related practices such as alcohol use.
Group 3, the contrast group, represents almost a fourth of the sample and it is defined by contrast in the CFNI subscales: high gender role conformity in most of the subscales except in involvement in children, sexual fidelity and modesty, and a medium score in domesticity. The group presents an external image adapted to femininity standards, but is more liberal in terms of sexual freedom, less focused on children and not much interested in showing their abilities.
Possible explanations for these findings could be related with the growing sector of Spanish women engaged in highly demanding professions, which force them to choose between involvement in family and career aspirations (Grau-Grau, 2013; Muñiz et al., 2014; Pérez-Caramés, 2014; Salido, 2002), thus developing more egalitarian gender roles and work-related attitudes and behaviors . This fact was observed in their patterns of alcohol use, more connected with the highly demanding professional and public life and less conservative households in which single or divorced women are more common (Domenico & Jones, 2006; Johnston, O’Malley, Bachman, & Schulenberg, 2011; Samet, Yoon, & Organization, 2010; Sánchez–López, et al., 2012). Nevertheless, social pressure in terms of physical appearance and kindness toward others is still a gender-related variable even among the most demanding and competitive job positions, at least in the Spanish context (Cuadrado, 2004; Ramírez, Escudero, & Arias, 2011). This hypothesis seems to be supported by the fact that a higher proportion of medium-high studies was found in Group 3, visualizing those younger Spanish women who, despite conforming to some gender roles, still try to be competitive in an increasingly demanding working environment. The same environment which demands feminine standards, such as being nice with others and preserving a great external appearance, also allows them to put in practice more liberal ideas and principles in their private life. They preserve their freedom and they choose to preserve their autonomy. They choose to keep single and they believe in sexual freedom more frequently than conservative groups do.
Finally, group 4, the non-conformist group, represents a small percentage of women who dismiss the social standards of feminine role conformity. These results coincide with other authors’ findings, that only a low proportion of women show low gender role conformity patterns (Mahalik et al., 2005; Sánchez-López, et al., 2009). Unexpectedly, a higher alcohol use was not reported among this group of women. Nevertheless, a higher presence of divorced, single or widowed women was found, concurring with the idea of marriage as reinforcement of gender inequalities in social structures (Chen, Fiske, & Lee, 2009; Sassler & Miller, 2011; Simon, 2002). This group of women, the youngest and with the highest educational level, seems to represent a growing sector of women that is more aware of gender inequalities. They struggle for their rights by adopting more equalitarian gender-related behaviors. Regarding alcohol use, previously linked to masculine habits(Sánchez–López, et al., 2012), it could be speculated that they do not need to adopt masculine habits to struggle for they rights.
--------Thanks for your comments and suggestions------
Artázcoz, L., Borrell, C., Rohlfs, I., Beni, C., Moncada, A., & Benach, J. (2001). Trabajo doméstico, género y salud en población ocupada. Gaceta Sanitaria, 15(2), 150-153.
Artazcoz, L., Escribà-Agüir, V., & Cortès, I. (2004). Género, trabajos y salud en España. Gaceta Sanitaria, 18, 24-35.
Borgonovi, F. (2012). The relationship between education and levels of trust and tolerance in Europe1. The British Journal of Sociology, 63(1), 146-167.
Carrasco, C., & Recio, A. (2001). Time, work and gender in Spain. Time & Society, 10(2-3), 277-301.
Cuadrado, I. (2004). Valores y rasgos estereotípicos de género de mujeres líderes. Psicothema, 16(2), 270-275.
Chen, Z., Fiske, S. T., & Lee, T. L. (2009). Ambivalent sexism and power-related gender-role ideology in marriage. Sex Roles, 60(11-12), 765-778.
Domenico, D. M., & Jones, K. H. (2006). Career aspirations of women in the 20th century. Journal of career and technical education, 22(2), n2.
Esteban-Gonzalo, S., Aparicio, M., & Estaban-Gonzalo, L. (2018). Employment status, gender and health in Spanish women. Women & health, 58(7), 744-758.
Flaquer, L., Mínguez, A. M., & López, T. C. (2016). Changing family models: Emerging new opportunities for fathers in Catalonia (Spain)? Balancing Work and Family in a Changing Society (pp. 65-81): Springer.
González, M. J., Jurado, T., & Naldini, M. (2014). Introduction: Interpreting the transformation of gender inequalities in Southern Europe Gender Inequalities in Southern Europe (pp. 10-40): Routledge.
Gracia, P. (2014). Fathers’ child care involvement and children’s age in Spain: A time use study on differences by education and mothers’ employment. European Sociological Review, 30(2), 137-150.
Grau-Grau, M. (2013). Clouds over Spain: Work and family in the age of austerity. International Journal of Sociology and Social Policy, 33(9/10), 579-593.
Guner, N., Kaya, E., & Sánchez-Marcos, V. (2014). Gender gaps in Spain: policies and outcomes over the last three decades. SERIEs, 5(1), 61-103.
Johnston, L. D., O’Malley, P. M., Bachman, J. G., & Schulenberg, J. E. (2011). Monitoring the Future national survey results on drug use, 1975-2010. Volume I: Secondary school students.
López-Sáez, M., Morales, J. F., & Lisbona, A. (2008). Evolution of gender stereotypes in Spain: Traits and roles. The Spanish Journal of Psychology, 11(2), 609-617.
Mahalik, J. R., Burns, S. M., & Syzdek, M. (2007). Masculinity and perceived normative health behaviors as predictors of men's health behaviors. Social science & medicine, 64(11), 2201-2209.
Mahalik, J. R., Morray, E. B., Coonerty-Femiano, A., Ludlow, L. H., Slattery, S. M., & Smiler, A. (2005). Development of the conformity to feminine norms inventory. Sex Roles, 52(7-8), 417-435.
Mínguez, A. M. (2010). Family and gender roles in Spain from a comparative perspective. European Societies, 12(1), 85-111.
Muñiz, J., Peña-Suárez, E., de la Roca, Y., Fonseca-Pedrero, E., Cabal, Á. L., & García-Cueto, E. (2014). Organizational climate in Spanish Public Health Services: Administration and Services Staff. International Journal of Clinical and Health Psychology, 14(2), 102-110.
Organization, W. H., & Unit, W. H. O. M. o. S. A. (2014). Global status report on alcohol and health, 2014: World Health Organization.
Pérez-Caramés, A. (2014). Family policies in Spain Handbook of family policies across the globe (pp. 175-194): Springer.
Ramírez, B. F., Escudero, E. B., & Arias, E. E. (2011). Desplazamiento y normalización del rechazo laboral hacia las mujeres por cuestiones de talla. Prisma Social: revista de investigación social(7), 14.
Salido, O. (2002). Women’s labour force participation in Spain. Universidad Complutense de Madrid. Unidad de Políticas Comparadas (CSIC).
Samet, J. M., Yoon, S.-Y., & Organization, W. H. (2010). Gender, women, and the tobacco epidemic.
Sánchez-López, M. P., Flores, I. C., Dresch, V., & Aparicio-Garcia, M. (2009). Conformity to feminine gender norms in the Spanish population. Social Behavior and Personality: an international journal, 37(9), 1171-1185.
Sánchez–López, M. d. P., Cuellar–Flores, I., & Dresch, V. (2012). The impact of gender roles on health. Women & health, 52(2), 182-196.
Sassler, S., & Miller, A. J. (2011). Waiting to be asked: Gender, power, and relationship progression among cohabiting couples. Journal of Family Issues, 32(4), 482-506.
Scott, J. (2006). Family and gender roles: how attitudes are changing. Arxius de Ciències Socials, 15, 143-154.
Simon, R. W. (2002). Revisiting the relationships among gender, marital status, and mental health. American journal of sociology, 107(4), 1065-1096.

Reviewer 2 Report
I have reviewed the article entitled “Understanding the meaning of
conformity to feminine norms in lifestyle habits and health: A cluster
analysis” which is interesting and publishable in the journal after
addressing following comments.
What type of research methods during data collection were adopted?
Authors must have to declare in the article. Moreover, what about the
consents of respondents. Did the authors record the consent of
respondents or not? Authors must have to write that if the interviews
were based on the inform consent.
Authors must have to write the implementations of logistic regression in
various field. Before starting the second paragraph of section 2.4. Data Analyses, this sentence should add “Logistic regression has implemented in various fields of study particularly, crops (Elahi et al. 2019a), livestock (Elahi et al. 2018a), and social sciences (Adepoju, 2008; Wauters et al., 2010; Hanim et al., 2017). Similarly, we have used logistic regression to analyse the relationships between gender role conformity and civil status, alcohol consumption, tobacco consumption and physical activity………”
Elahi, E., Abid, M., Zhang, H., Weijun, C., & Hasson, S. U. (2018a).
Domestic water buffaloes: access to surface water, disease prevalence
and associated economic losses. Preventive Veterinary Medicine.
doi:org/10.1016/j.prevetmed.2018.03.021
Elahi, E., Khalid, Z., Weijun, C., & Zhang, H. (2019a). The public
policy of agricultural land allotment to agrarians and its impact on
crop productivity in Punjab province of Pakistan. Land Use Policy,
doi:org/10.1016/j.landusepol.2019.104324
Adepoju, A., 2008. Technical efficiency of egg production in Osun State. International Journal of Agricultural Economics and rural development 1, 7-14.
Hanim, A., Razman, M., Jamalludin, A., Nasreen, E., Phyu, H.M., SweSwe, L., Hafizah, P., 2017. Knowledge, Attitude and Practice on Dengue among Adult Population in Felda Sungai Pancing Timur, Kuantan, Pahang. International Medical Journal Malaysia 16.
Wauters, E., Bielders, C., Poesen, J., Govers, G., Mathijs, E., 2010. Adoption of soil conservation practices in Belgium: an examination of the theory of planned behaviour in the agri-environmental domain. Land use policy 27, 86-94.
Author Response
Dr Sara Esteban Gonzalo
Associate Professor
Faculty of Biomedical and Health Science
Universidad Europea
Villaviciosa de Odón, 28670 Madrid
Sara.esteban@universidadeuropea.es
05 February 2020
Dear Reviewer
RE submission of manuscript: No. ijerph-684541
Thank-you for giving us the opportunity to re-submit to International Journal of Environmental Research and Public Health a revised version of the manuscript entitled " Understanding the meaning of conformity to feminine norms in lifestyle habits and health: A cluster analysis.” by Esteban et al. The authors thank the reviewer for their thoughtful comments and suggestions. We have been able to incorporate all of the feedback and feel that the manuscript is stronger as a result.
All the authors have read and approved the final manuscript.
We look forward to your ongoing correspondence regarding our submission.
Sincerely,
Dr S.E.
Corresponding Author
Reviewer 2.
I have reviewed the article entitled “Understanding the meaning of
conformity to feminine norms in lifestyle habits and health: A cluster
analysis” which is interesting and publishable in the journal after
addressing following comments.
Thank you so much for this encouraging comments.
What type of research methods during data collection were adopted?
Authors must have to declare in the article. Moreover, what about the
consents of respondents. Did the authors record the consent of
respondents or not? Authors must have to write that if the interviews
were based on the inform consent.
Thank you for this relevant comment. Information related to data collection was scarce. We have included extra information, in order to address the reviewer’s comment.
The written consent was obtained from each of the participant. We have also included this information in the method section.
2.1. Participants
The study sample was composed of 347 women. All participants were currently living in Spain; most of them were Spanish women (93.7%), age 18-70 (mean=42.2 years). Data collection was conducted during 2014 with the collaboration of several organizations from various sectors (health care, telecommunications, education, engineering, banking and insurance, business administration, and marketing and design), local employment agencies, and with local women’s associations in more rural areas. These organizations offered the opportunity to their employees or associated members to participate voluntarily in this study.
The eligibility criterion was to be employed by or a member of one of the cooperating organizations. Ninety percent of women agreed to participate in the study, and around 80% of questionnaires were fully answered and their data retained for the analyses. Missing answers were particularly observed in sexual content items among older women. A total of 86 (20%) questionnaires with missing answers were discarded. Data collection was conducted during 2014 and the data collection process was completed in approximately 6 months. All women received written questionnaires to answer and return to the researcher, in which they were requested to affirm that they met inclusion criteria (which was also confirmed by study staff). The average time needed to answer the questionnaire was 45-60 minutes after prior clarification of instructions. The participants were supported at all times by a person trained to answer their questions whenever necessary, especially regarding older women. All participants were informed of the purpose and intent of the study and provided written consent. Similarly, anonymity of each of the participants was ensured.
Authors must have to write the implementations of logistic regression in
various field. Before starting the second paragraph of section 2.4. Data Analyses, this sentence should add “Logistic regression has implemented in various fields of study particularly, crops (Elahi et al. 2019a), livestock (Elahi et al. 2018a), and social sciences (Adepoju, 2008; Wauters et al., 2010; Hanim et al., 2017). Similarly, we have used logistic regression to analyse the relationships between gender role conformity and civil status, alcohol consumption, tobacco consumption and physical activity………”
Elahi, E., Abid, M., Zhang, H., Weijun, C., & Hasson, S. U. (2018a).
Domestic water buffaloes: access to surface water, disease prevalence
and associated economic losses. Preventive Veterinary Medicine.
doi:org/10.1016/j.prevetmed.2018.03.021
Elahi, E., Khalid, Z., Weijun, C., & Zhang, H. (2019a). The public
policy of agricultural land allotment to agrarians and its impact on
crop productivity in Punjab province of Pakistan. Land Use Policy,
doi:org/10.1016/j.landusepol.2019.104324
Adepoju, A., 2008. Technical efficiency of egg production in Osun State. International Journal of Agricultural Economics and rural development 1, 7-14.
Hanim, A., Razman, M., Jamalludin, A., Nasreen, E., Phyu, H.M., SweSwe, L., Hafizah, P., 2017. Knowledge, Attitude and Practice on Dengue among Adult Population in Felda Sungai Pancing Timur, Kuantan, Pahang. International Medical Journal Malaysia 16.
Wauters, E., Bielders, C., Poesen, J., Govers, G., Mathijs, E., 2010. Adoption of soil conservation practices in Belgium: an examination of the theory of planned behaviour in the agri-environmental domain. Land use policy 27, 86-94.
Thank you very much for this suggestion. We have modified the paragraph as you proposed us, adding as well the cited references.
Logistic regression has been implemented in various fields of study, particularly in research into crops[25]and livestock[26], and in the social sciences[27-29]. Binary logistic regression models were constructed to analyze the relationships between gender role conformity and civil status, alcohol consumption, tobacco consumption and physical activity, controlling for confounding variables (age and educational level). In addition, linear regression models were built to assess the relationship between gender role conformity and sleep duration and social support.
--------Thanks for your comments and suggestions------
